# Chromosome Microarray Analysis and Exome Sequencing: Implementation in Prenatal Diagnosis of Fetuses with Digestive System Malformations

**DOI:** 10.3390/genes14101872

**Published:** 2023-09-26

**Authors:** You Wang, Liyuan Liu, Fang Fu, Ru Li, Tingying Lei, Ruibin Huang, Dongzhi Li, Can Liao

**Affiliations:** 1The First School of Clinical Medicine, Southern Medical University, Guangzhou 510515, China; wy13781539630@163.com (Y.W.); 18203186128@163.com (L.L.); 2Department of Prenatal Diagnostic Center, Guangzhou Women and Children’s Medical Center, Guangzhou Medical University, Guangzhou 510620, China; fuyingyi2008@163.com (F.F.); linra020@126.com (R.L.); leiyuru@126.com (T.L.); huangruibin96@hotmail.com (R.H.); lidongzhi2013@aliyun.com (D.L.)

**Keywords:** digestive system malformations, prenatal diagnosis, exome sequencing, chromosome microarray analysis, genetic variants

## Abstract

(1) Purpose: Retrospective back-to-back comparisons were performed to evaluate the accuracy, effectiveness, and incremental yield of chromosome microarray analysis (CMA) and exome sequencing (ES) analysis in fetuses with digestive system malformations (DSMs). (2) Methods: In total, 595 women with fetal DSMs who underwent prenatal diagnosis were enrolled. We analyzed the diagnostic yields of CMA and ES and evaluated pregnancy outcomes. Copy number variants (CNVs) were classified according to the American College of Medical Genetics and Genomics guidelines. (3) Results: Pathogenic CNVs were detected in 11/517 (2.12%) fetuses, and variants of unknown significance (VUS) were identified in 69 (13.35%) fetuses using CMA. ES detected 29 pathogenic/likely pathogenic variants in 23/143 (16.08%) fetuses and 26/143 (18.2%) VUS. In those with other ultrasound abnormalities, the detection rate of multiple system structural malformations was 41.2%, followed by skeletal (33.3%), cardiovascular (25.4%), and central nervous system (18.6%) malformations. Of the 391 surviving children, 40 (10.2%) exhibited varying degrees of mental retardation. (4) Conclusion: A correlation exists between DSMs and chromosomal abnormalities. When combined with other systemic abnormalities, the incidence of chromosomal abnormalities increases significantly. Patients with congenital DSM are at risk of developing neurodevelopmental disorders. Combined CMA and ES detection of fetal DSM has good clinical application potential.

## 1. Introduction

Fetal digestive system malformation (DSM) is a common congenital malformation, accounting for approximately 7.35% of all fetal malformations [1]. Its incidence is second only to cardiovascular and skeletal malformations, with a total prevalence of approximately 15/10,000 European births per year [2,3]. The clinical manifestations of DSM differ, and the severity of different disease subgroups varies. However, most cases are characterized by vomiting, abnormal defecation, and feeding difficulties after birth and can be life-threatening, with severe cases often requiring surgical treatment [2]. The prenatal diagnosis and classification of DSMs rely mainly on prenatal ultrasound [4]. Typical DSM diseases include omphalocele, esophageal atresia, Hirschsprung disease, intestinal atresia, gastroschisis, and anorectal malformations. Congenital DSMs are caused by genetic or environmental factors, the interaction between these two factors, or other unknown causes. Current research shows that DSMs are closely related to chromosomal abnormalities [5,6]. According to the United States statistics of birth defects, the common chromosomal abnormalities in gastrointestinal malformations are trisomy 21, 18, and 13 and Turner syndrome [7]. Abnormal expressions of genes such as *HOX* and *FGF* may also lead to DSMs [8,9].

Fetal structural abnormalities are the main indications for invasive prenatal genetic testing, which is traditionally performed using karyotype analysis. The incidence of chromosomal abnormalities is approximately 5.4–15.5% when ultrasound identifies multiple system abnormalities in a fetus [10,11]. Further, chromosome microarray analysis (CMA) detection of fetuses with a negative karyotype results in an increase in the detection rate of genomic microdeletions and microduplications by 4–6% [12,13,14]. Nevertheless, the underlying causes of chromosomal abnormalities in most fetuses with structural abnormalities remain unclear. Recently, with the rapid development of next-generation sequencing technology, exome sequencing (ES) has been used as a diagnostic tool to assess postpartum patients with previously undiagnosed suspected genetic diseases, with an average molecular diagnostic rate of 25% [15,16,17]. In addition, this technology is gradually being applied to prenatal diagnosis [15]. ES increased the diagnostic rate by 8.5–11.6% for fetuses with abnormal ultrasound but negative karyotype and CMA results [18,19,20]. However, although many studies show that fetal DSMs have a higher risk of various genetic diseases and syndromes, few studies used CMA and ES detection techniques to elucidate the phenotype–genotype correlation [5,6,7,8].

In this study, we aimed to evaluate the detection efficacy of microscopic and submicroscopic chromosomal abnormalities and single-gene variants in fetuses with DSMs and evaluate perinatal prognosis to provide more accurate information for parental counseling.

## 2. Materials and Methods

### 2.1. Participant Recruitment

This retrospective study reviewed clinical details, genetic diagnoses, and pregnancy outcomes of patients with prenatal digestive system abnormalities. This study was approved by the Ethics Committee of the Guangzhou Women and Children’s Medical Center, and the participants provided written informed consent. We recruited 595 women carrying fetuses with DSMs confirmed by prenatal ultrasound or magnetic resonance imaging (MRI), with/without other structural abnormalities. This study excluded patients with congenital diaphragmatic hernia because its pulmonary comorbidities may require extracorporeal membrane oxygenation.

All cases were divided into isolated and nonisolated groups based on whether the DSM was combined with other structural abnormalities. In this study, the fetus was classified into the isolated group regardless of whether the fetus had ultrasound soft markers or not. Ultrasonic soft markers included absent or hypoplastic nasal bone, choroid plexus cysts, thickened nuchal folds, a single umbilical artery, a persistent left superior vena cava, an echogenic intracardiac focus, and an echogenic bowel. All patients consulted two or more professional ultrasound doctors. The parents underwent invasive prenatal diagnosis at our prenatal diagnosis center. All the clinical data of the patients were obtained through our center’s ultrasound system and medical record database. All parents underwent regular physical examinations and physical health tests, and there were no blood relationships. Comprehensive genetic counseling was provided for pregnant women and couples, which included information about the potential benefits and risks associated with each invasive procedure.

### 2.2. Sample Collection

Fetal samples were obtained via amniocentesis (gestational age <25 weeks) and cord blood sampling (>25 weeks or oligohydramnios). Using a QIAamp DNA Blood Midi/Mini Kit (Qiagen GmbH, Dusseldorf, Germany), fetal genomic DNA samples were obtained from 15 mL of amniotic fluid or 2 mL of cord blood, following the manufacturer’s instructions. Parental blood samples for trio-ES analysis were collected with informed consent.

### 2.3. Genetic Testing Tools

#### 2.3.1. Karyotype Analysis

Quantitative fluorescence polymerase chain reaction was used to rapidly diagnose common aneuploidy prenatally (13, 18, 21, X, or Y) in all 595 fetuses with DSM. Standard G-banded karyotype analysis (550-band resolution) was used to identify overall chromosomal abnormalities. The karyotype was named according to the International System for Human Cytogenomic Nomenclature 2015 guidelines.

#### 2.3.2. Chromosomal Microarray Analysis

We performed whole-genome high-resolution microarray analysis with CytoScan 750 K and CytoScan HD arrays (Affymetrix, Santa Clara, CA, USA) to investigate submicroscopic genomic imbalances in fetal samples with normal karyotypes. The data detected using the CHAS 3.1 software supporting the kit were analyzed, and the results of the single nucleotide polymorphism (SNP) array were analyzed in combination with the relevant database to determine the nature of copy number variations (CNVs). The American College of Medical Genetics (ACMG) guidelines were followed for data analysis, which classified all identified CNVs as pathogenic (P), likely pathogenic (LP), variants of uncertain significance (VUS), likely benign, or benign. The referenced databases (Appendix A) included internal databases of sample laboratories and online international public databases. All de novo CNVs identified using CMA were verified using secondary techniques, such as quantitative real-time polymerase chain reaction or fluorescence in situ hybridization. Additionally, CMA was performed on DNA samples isolated from the parental whole blood to aid in interpreting CNVs and identifying inheritance patterns.

#### 2.3.3. Exome Sequencing

ES includes whole-exome sequencing (WES) and clinical exome sequencing. Trio-ES testing was performed on the parents and fetuses with negative karyotyping and CMA results. Following the manufacturer’s instructions, DNA libraries were generated using a NEXTflexTM Rapid DNA Sequencing Kit (5144-02) (Bioo Scientific, Austin, TX, USA). The DNA samples were analyzed using an Agilent Bioanalyzer 2100 (Agilent Technologies, Santa Clara, CA, USA) in accordance with the manufacturer’s instructions. Samples were sequenced (the average depth was above 100×, and the coverage was above 99.75%) using a HiSeq2500 sequencer (version 3; Illumina, Inc., San Diego, CA, USA). Paired-end sequencing was performed for each sample.

Appendix A describes the ES data analysis and interpretation. Illumina bcl2fastq software (version 3) was used to process the raw images for base calling and raw data generation. Trimmomatic was used to filter adapter-contaminated and low-quality readings (quality threshold: 20). The Burrows–Wheeler method was used to align these readings with the NCBI human reference genome (hg19/GRCh37). Using SAMtools and Pindel, the BAM files were subjected to SNP analysis, duplication marking, indel realignment, and recalibration. The minor allele frequencies of all known variants were also reported based on their presence in the dbSNP (http://www.ncbi.nlm.nih.gov/snp, accessed on 23 January 2023), 1000 Genomes Project (1000 GP) (http://browser.1000genomes.org, accessed on 26 January 2023), Exome Variant Server (http://evs.gs.washington.edu/EVS, accessed on 23 January 2023), and Exome Aggregation Consortium (http://exac.broadinstitute.org/, accessed on 26 January 2023) databases. Where appropriate, databases (Appendix A) were used to determine mutation harmfulness and pathogenicity. Analyses of biological effects were performed on all whole-exome variations, which included the use of programs (Appendix A) to assess whether an indel or amino acid substitution would have a significant biological impact.

Data filtering and analysis have been described in detail in our previous studies. According to the ACMG guidelines, all selected variants were classified as P, LP, or VUS and likely benign or benign.

All reported variants were verified by Sanger sequencing. Our study’s sample-to-result turnaround time was approximately 3–8 weeks.

### 2.4. Intelligence and Growth Assessment

The assessment of children’s growth and development included the National Center for Health Statistics standard recommended by the World Health Organization as the evaluation standard, and age-for-height, age-for-weight, and height-for-weight were used as evaluation indices for height, weight, growth, and development. This evaluation was conducted in accordance with previous studies [21]. The weight-for-age Z score, height-for-age Z score, and weight-for-height Z score were calculated using Anthro software (version 2.0). The established tools and their details are shown in Appendix A.

Neurodevelopmental evaluations included the assessment of physical, motor, sensory, and cognitive development in infants and young children and the measurement of intelligence quotient or adaptive level of functioning among older children. Neurodevelopmental assessments were performed as described in previous studies [22,23]. Specific evaluation procedures for the nervous system were described in detail in our previous studies [24]. According to the latest assessment criteria for mental retardation, we divided patients into three levels: intellectual disability (<70), borderline intelligence (70–84), and normal intelligence (≥85) [22,25]. The ophthalmologic evaluation included external ocular examination, indirect ophthalmoscopy, and measurement of the best-corrected visual acuity (Snellen score) in children [24].

### 2.5. Statistical Analysis

All statistical analyses were performed using IBM Statistical Program SPSS 26.0 (IBM, Armonk, NY, USA). Statistical significance was set as a *p*-value < 0.05.

### 2.6. Clinical Follow-Ups

The hospital’s internal clinical information registration system and telephone follow-up were used to obtain late fetal ultrasound review results, pregnancy outcomes, and fetal growth and development after birth.

## 3. Results

### 3.1. Cohort Characteristics

In total, 595 pregnant women diagnosed with fetal DSMs using prenatal ultrasound/MRI underwent invasive surgery for prenatal diagnosis at our prenatal diagnosis center between 1 January 2013 and 1 January 2023. A progress flowchart of the prenatal diagnosis analysis is shown in Figure 1. The gestational age distribution of prenatal diagnosis was 13 + 3–36 + 1 weeks, with an average gestational age of 25.14 ± 4.05 weeks. Of these, 443 cases (74.45%) underwent amniocentesis and 152 (25.55%) underwent cord blood puncture. Among these, 348 (58.49%) were male and 247 (41.51%) were female. According to whether other ultrasound abnormalities were present, the cases were divided into an isolated group (492 cases, 82.69%) and a nonisolated group (103 cases, 17.31%). In this study, fetal DSMs included 17 diseases. The top 10 diseases were duodenal obstruction, absent gallbladder, abnormal gastric vesicles, meconium peritonitis (each patient showed one or more of the following ultrasound findings: intra-abdominal calcification or echogenic mass, fetal ascites, polyhydramnios, dilated bowel loop, etc.), hepatobiliary abnormalities, small intestinal obstruction, intestinal duplication, esophageal atresia, hepatic hemangioma, and intrahepatic calcification, accounting for 82.01% (488/595) of the cases in this study (Appendix A). There were 103 cases (17.31%) of fetal DSMs combined with other ultrasound abnormalities, of which amniotic fluid volume abnormalities were the most common, accounting for 45.63% (47/103) (including 38 cases of polyhydramnios, 8 of oligohydramnios, and 1 without amniotic fluid).

### 3.2. Karyotype Analysis and CMA Results

Overall, 2.52% (15/595) of fetuses had abnormal karyotypes. Trisomy 21 was the most common abnormality, detected in eight samples (1.35%). CMA was successfully performed in 517 fetuses with DSMs, including 93 in the nonisolated group and 424 in the isolated group. CNVs with clinical significance were identified in 11 fetuses by CMA, with a total detection rate of 2.12% (11/517). The detection rates of pathogenic CNVs were 2.15% (2/93) in the nonisolated group and 2.12% (9/424) in the isolated group. VUS were found in 69/517 (13.35%) fetuses, and 73.9% (51/69) of VUS originated from the parents. Table 1 summarizes the LP and P CNVs identified by CMA in our cohort.

### 3.3. Exome Sequencing Results

#### 3.3.1. Positive Diagnostic Variations

Trio-based WES was performed on 143 pairs of parents and two sisters (a total of 431 people, including father–mother–proband–fetal samples) of 143 fetuses with negative karyotypes and negative CMA results or VUS. The coding regions targeted by ES had a mean coverage of 102×. In the targeted coding regions, a mean of 99.56% of bases were covered by at least 10 reads. CNVs in all samples were evaluated using high-resolution microarray analysis, with a minimum resolution of up to 25–50 kb; therefore, we did not pay special attention to CNV analysis in the ES data. Overall, ES detected 29 P/LP variants, 3 cases of microdeletions, microduplications in 23 cases (Table 2), and 26 VUS in 26 cases (Appendix A). Of the 29 diagnostic variants identified, 14 variants were not reported previously. The diagnostic yield of ES was 16.08% (23/143).

In the 23 positive cases, de novo variants were identified in 11 cases. In these 11 cases, 10 cases (90.9%) were autosomal dominant and 1 case (9.1%) was autosomal recessive. Compound heterozygotes or homozygotes were found in 10 cases with autosomal recessive inheritance. One hemizygote was inherited from his mother with X-linked dominant/recessive inheritance. The remaining fetus inherited from his mother without a clinical phenotype with autosomal dominant.

The following diseases were found in two or more cases: Joubert syndrome (IFT74/CPLANE1, n = 2) and Noonan syndrome (PTPN11/KRAS, n = 3).

#### 3.3.2. Diagnostic Rate of DSMs in Different Subgroups

Table 3 summarizes the diagnostic rates of the different subgroups of fetal DSMs in this study. The detection rate of ascites was the highest (n = 10/32, 31.25%), followed by abdominal masses (n = 1/4, 25.0%) and hepatomegaly (n = 4/18, 22.2%). The diagnostic rates of the isolated DSMs (4.3%, 2/46) and nonisolated DSMs groups (21.6%, 21/97) were significantly different (*p* = 0.007) (Table 4). The detection rate of DSM with multiple system malformations was the highest (41.2%), followed by skeletal system (33.3%), cardiovascular system (25.4%), central nervous system (18.6%), and urinary system (10.3%) malformations.

#### 3.3.3. Variants of Unknown Significance

VUS (18.2%, 26/143) were detected in 26 fetuses with DSMs (Appendix A), including 5 cases of X-linked recessive/dominant inheritance, 3 cases of X-linked recessive inheritance, 4 cases of autosomal recessive/dominant inheritance, 10 cases of autosomal dominant inheritance, and 4 cases of autosomal recessive inheritance. Of these, 88.5% (23/26) of VUS originated from the parents. The detection rates of VUS in the nonisolated group (26.8%) and the isolated group (2.2%) were significantly different (*p* < 0.05).

### 3.4. Pregnancy Outcomes

We obtained information on perinatal outcomes of 557 (93.61%) pregnancies (Table 4); 38 patients were lost to follow-up. Among the successful follow-up cases, 149 were terminated, 391 were live births, 27 were premature births, 32 had a low birth weight, the survival rate was 70.20%, 11 were intrauterine deaths, and 6 were neonatal deaths. Among the live births, 334 (85.42%) received postpartum ultrasound reexamination and 57 (14.58%) did not. We compared the postpartum phenotype with prenatal ultrasound findings by using HPO terms. In total, 308 cases (78.77%) were found to be consistent with prenatal and postpartum ultrasound findings. A total of 26 cases (6.65%) showed new phenotypes in postpartum ultrasound scans, including 7 cases of atrial septal defect, 5 cases of ventricular septal defect, 5 cases of renal cyst, 4 cases of biliary atresia, 4 cases of congenital clubfoot, and 1 case of renal agenesis. Among the 11 cases with positive CMA results, 9 terminated the pregnancy, 1 had a live birth, and there was 1 neonatal death. Among the 23 cases with positive ES results, 13 terminated the pregnancy, 9 were live births, and there was 1 neonatal death. There were 85 cases with VUS (10 with both CMA and ES with VUS). Among the cases with VUS, 19 were terminated, 61 (71.7%) were live births, 3 were neonatal deaths, and 2 were lost to follow-up. A total of 238 patients were transferred for neonatal or pediatric surgery after birth. The pediatric hospitalization rate was 60.87% (238/391) and the operation rate was 40.15% (157/391). The results of the unconditional logistic regression analysis showed that combined extradigestive system abnormalities were risk factors for adverse pregnancy outcomes (termination of pregnancy/stillbirth). The risk of adverse pregnancy outcomes in fetuses with extradigestive malformations was 13.14 times higher than that in the isolated group (95% confidence interval: 5.16–33.47).

### 3.5. Results of Growth and Intelligence Assessment

Through the follow-up of 391 surviving fetuses, we found that 10 with positive CMA and ES test results were evaluated for growth and development. Three patients (30%) showed growth retardation with Z scores of −2, −1, and −1, respectively. Three patients (30%) showed mental retardation with intelligence scores of 61, 66, and 70, respectively. Among the 61 surviving fetuses with VUS, 41 were confirmed to have DSMs after birth and underwent surgical treatment within 3–47 months of birth. The length of hospital stay ranged from 7 to 21 days. Of the surviving infants, 9.8% (6/61) had growth retardation, 16.4% (10/61) had mild intellectual disability, 1.6% (1/61) had mild to severe intellectual disability, and 3.27% (2/61) had mild visual impairment. The Z scores of the six children with growth retardation were −4, −3, −2, −1, −1, and 0, respectively. The median age at the last assessment was 5.2 years (range: 1.9–10.1 years) in patients who underwent an intelligence assessment. The intelligence scores of the five children with mental retardation were 66, 69, 70, 72, and 74. The average language score was 66 (range 53–88). Among the 323 surviving children with negative genetic test results, 31 (9.6%) had different degrees of growth retardation, with an average Z score of −1 (−4–1), and 26 children (8.0%) had varying degrees of mental retardation, with a median score of 56 (41–77). Mild to moderate visual problems were observed in 14 patients (4.3%). Among 14 cases of short bowel syndrome in the 391 surviving fetuses, 5 cases (35.7%) had different degrees of mental retardation, including 1 case of moderate to severe with an intelligence score of 56. 

The incidence of mental retardation was not statistically different between the negative and positive groups in the genetic test results but was significantly different from that in the VUS group (*p* = 0.047 and 0.010, respectively). There was no significant difference in the incidence of growth retardation between the negative and positive groups and the VUS group (*p* = 0.071 and 0.860, respectively).

## 4. Discussion

Compared with traditional karyotype analysis, CMA and ES detection techniques allow the identification of other clinically relevant chromosomal abnormalities and increase diagnostic yield [18,19,20,26,27]. Previous studies on chromosomal abnormalities in DSMs have limited data, most of which are limited to traditional chromosomal karyotype analysis and are mainly based on disease subgroups of DSMs, with a low diagnostic yield [28]. In this study, we performed a detailed prenatal genetic etiology analysis of 595 fetuses with DSMs in a large cohort. We not only evaluated the diagnostic yield of CMA and ES but also followed up on their clinical prognosis. This is, by far, the largest prenatal study of DSM to date.

Overall, our results showed that CMA provided a diagnostic rate of 2.12% for cases with negative karyotype results, and ES provided an additional diagnostic rate of 16.08% for cases with negative karyotype and CMA results. The detection rate of CMA was slightly lower than that observed in previous studies, and the detection rate of ES was slightly higher, which may be related to sample selection bias. In our study cohort, the detection rate of CMA was 2.12%, and there was no significant difference between the isolated and nonisolated groups, demonstrating that CNV detection should not be limited to nonisolated DSMs. 

Standard G-banding can detect approximately 1/3 of chromosomal abnormalities, but karyotype analysis still misses CNVs of 5 Mb size, suggesting that most families should choose the CMA or CMA + karyotype instead of karyotype analysis alone. In addition, the advantage of implementing CMA + karyotype compared to only implementing CMA is that when routine chromosome karyotype analysis is performed for any indication, the detection rate of abnormalities is 0.6% higher than CMA. However, as we and others have shown, most of this rate can be explained by balanced inversions and translocations. Triploids can also be detected through routine chromosome karyotype analysis but missed diagnosis by CMA [13]. The additional detection rate of trio-ES for negative karyotypes and CMA cases was 16.08%, suggesting that CMA combined with trio-ES may be an effective method for the diagnosis of the genetic etiology of DSM, which is worth recommending. In addition, the importance of analyzing trios to improve the diagnostic rate of ES has been well demonstrated, both after birth and prenatally [29,30]. Among the 23 positive cases in this study, only 47.8% of the diagnostic variants were de novo variants with dominant inheritance. These data are slightly lower than those of previous studies in our unit [31], which may be related to the small number of cases of trio-ES in this study; however, whether it is related to the hereditary characteristics of the DSM disease itself cannot be ruled out. We believe this issue requires further investigation. 

Our study also found that the genetic diagnostic yield of fetuses with DSM with multiple structural abnormalities was significantly higher than that of isolated DSM fetuses (21.6% vs. 4.3%, *p* = 0.007), which is consistent with previous studies [30]. It is worth noting that the classification of isolated and nonisolated DSM was based on the results of prenatal ultrasonography in this study. Although we have corrected this shortcoming by using HPO terms regarding the postnatal phenotype and comparing it to the prenatal ultrasound findings, in fact, fetal phenotypes that can be identified using prenatal ultrasound are limited. For example, mental retardation may be unrecognizable on prenatal ultrasonography. These objective factors complicate the prenatal ES analysis.

In our study, the detection rate of CMA was lower than that reported in previous studies, which may be related to the intervention for severe structural malformations detected using ultrasound during the first trimester of pregnancy. Most fetal DSMs are diagnosed during the second or third trimester [32]. The average gestational age at prenatal diagnosis in this study was 25.14 weeks, indicating that other major structural and aneuploidy abnormalities were excluded to some extent before this gestational age. Previous studies have shown that the detection rate of ES in fetuses with skeletal abnormalities is relatively high. Our study confirms this hypothesis. In this study, the detection rate of ES in DSMs combined with skeletal system malformations was as high as 33.3%, second only to the combination of multiple system malformations (41.2%). Our study also showed that the detection rate of ES in DSM combined with central nervous system malformations was 18.6%. This result indicates that when the fetus is diagnosed with DSM before delivery, further molecular diagnosis after routine testing (karyotype analysis and CMA) is particularly important, as it can identify diseases involving nervous system phenotypes that prenatal couples are most concerned about. In summary, considering that DSMs (isolated and nonisolated) have a high detection rate, we recommend that invasive prenatal diagnosis be considered for all pregnancies with DSMs detected by ultrasound, and genomic analysis should be performed for these cases.

Studies on pregnancies with suspected chromosomal abnormalities showed that 1.6–5.3% of fetuses carry P/LP CNVs and 1.4–5.2% of fetuses carry VUS [13,33,34,35,36,37]. In our study, the detection rates of VUS was 13.35% by CMA and 18.2% by ES. The detection rate of VUS using both methods was higher than that reported in previous studies (1.4–5.2%) [13,33,34]. This may be related to the different case groups that we selected and the fact that some studies included a normal population. Studies have shown that most VUS are hereditary [36,38]. Our research confirms this finding. In this study, 73.9% of CMA detections and 88.5% of ES detections originated from parents. Some reports showed that studies of parental origin can be used to check VUS and assist in the clinical interpretation of fetal CNVs and possible pregnancy outcomes [36,39,40].

In this study, we conducted a detailed clinical study of all 74 parents carrying fetal VUS but did not find any phenotypic characteristics of potential genetic diseases, indicating that VUS either have no disease association or have a subtle or variable disease penetrance. Furthermore, our study suggests that not only prenatal diagnosis results but also prenatal ultrasound results are important factors for couples to consider regarding pregnancy outcomes. It is undeniable that some VUS may be polymorphic genetic variants that are not associated with major ultrasound abnormalities; however, pregnancy-related decisions depend largely on ultrasound results. Therefore, we suggest that when abnormal signs of DSM are found in the fetus, other systems should be carefully scanned repeatedly, and dynamic monitoring should be performed for abnormal findings that cannot be confirmed. When fetal esophageal atresia, intestinal obstruction, intestinal malrotation, meconium peritonitis, or meconium pseudocysts are suspected, further fetal MRI examination may be considered to assist in the diagnosis [41].

Our study found that only a small number of pregnant women with hereditary VUS (8/74, 10.8%) chose to terminate the pregnancy. On the contrary, 11 out of 21 de novo VUS cases (52.4%) chose to terminate pregnancy. The mean duration of follow-up time was 5.1 years. Among the 68 surviving children with VUS, 9.8% had growth retardation, 18.05% had varying degrees of mental retardation, and 3.27% had mild visual impairments. We do not believe this was coincidental. Some cohort studies showed that the vast majority of CNVs classified as VUS may have nondisease outcomes in newborns [36,39]. This conclusion is not consistent with that of our study, and the source of the difference may be the difference in the length of follow-up and the different populations of the included cases. This difference may also confirm the variable penetrance of the disease to a certain extent, which undoubtedly increases the difficulty for clinicians in providing genetic counseling for such cases. Kong et al. [38] proposed that even VUS inherited from the parent can show variable penetrance when transmitted to the fetus, resulting in a disease phenotype in the later stages after birth, and should be followed up for a long time. Our study confirms this finding. Our study further emphasizes that special attention should be paid to cases with prenatal diagnostic results rated as VUS, whether from the parents or de novo. Clinicians and genetic counselors should obtain as much current information as possible to provide couples with the most detailed genetic counseling to help them make the best clinical decisions.

A recent review of 47 studies on neurodevelopmental outcomes in patients with congenital gastrointestinal malformations suggested that overall neurodevelopmental outcomes in these patients were significantly worse than those in patients with normal data or healthy controls [3]. Similar results were obtained in the present cohort where 6.7% of children with DSMs had different degrees of mental retardation, and the overall neurodevelopmental results were poor, especially in cases with positive diagnostic results or VUS. Roorda et al. [3] showed that children with DSMs are at an increased risk of neurodevelopmental disorders. Our study confirms this finding. Our study also showed that the incidence of mental retardation in children with DSM with VUS was significantly higher than those with negative results (18.0% vs. 8.0%, *p* = 0.010). This result indicates that clinicians should pay attention not only to clear positive results but also to cases rated as VUS when providing genetic counseling to pregnant couples. Some studies suggest that for children with DSMs, a longer average hospital stay and a higher average number of operations are associated with more severe overall neurodevelopmental disorders and impaired motor development. A similar trend was observed in the present study. This suggests that the more complex the course of the disease and/or treatment required in children with DSMs, the more profound the impact on neurodevelopmental outcomes. Therefore, early and accurate diagnosis of fetuses with DSM, identification of potential causes including genetic factors, and early intervention may be helpful for a good prognosis. In summary, our study showed that patients with congenital DSMs are at risk of small to medium damage in terms of neurodevelopmental outcomes.

The advantage of this study is that prenatal fetal DSMs were studied using CMA and trio-based ES in a large cohort, and the diagnostic rates and survival prognoses between the different DSM phenotypic subgroups were analyzed in detail. Our study has some limitations as well. First, this was a retrospective study; there may be memory bias, and it was difficult to obtain fresh blood from patients for cell or protein chemistry experiments. Second, it was not a continuous series, and only fetuses with qualified genomic DNA for CMA and ES analysis and availability of both parental samples may have created selection bias. Furthermore, although we mentioned in our study that the incidence of mental retardation in DSM children with VUS is significantly higher than those with negative results, we cannot deny that the true significance of VUS must be confirmed by other studies. Finally, owing to the limitations of sequencing methods, some variants could not be reliably detected.

## 5. Conclusions

Overall, our study revealed a correlation between DSMs and chromosomal abnormalities. When combined with other systemic abnormalities, the incidence of chromosomal abnormalities increases significantly. Conventional karyotype analysis combined with CMA detection can significantly improve the pathogenic detection rate in fetuses with DSM. ES technology can assist CMA in improving the molecular diagnostic rate of fetal structural abnormalities. When prenatal ultrasound examination revealed multiple fetal malformations and negative chromosome karyotype and/or CMA results, ES was recommended. The detection rates of chromosomal abnormalities in different types of DSMs vary, and the detection rate of chromosomal abnormalities in fetuses with ascites combined with other structural malformations was the highest. We speculate that CMA and ES combined with the detection of fetal DSMs is feasible and has good clinical application value.

## Figures and Tables

**Figure 1 genes-14-01872-f001:**
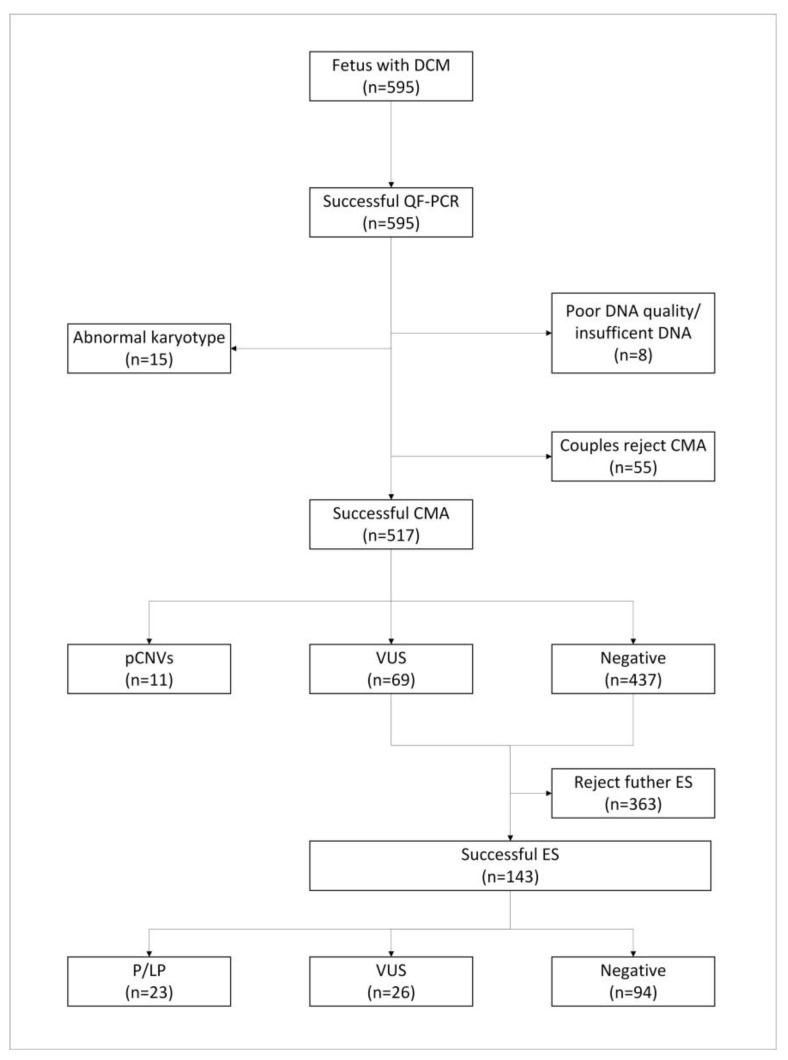
Flowchart of genetic analysis progression in a cohort of DSM fetuses. DSM: digestive system malformation; QF-PCR: quantitative fluorescent polymerase chain reaction; CMA: chromosomal microarray analysis; CNVs: copy number variants; VUS: variants of unknown significance; ES: exome sequencing; P: pathogenic; LP: likely pathogenic.

**Table 1 genes-14-01872-t001:** Clinically relevant characteristics of fetuses with DSM and clinically significant CMA findings.

Patient ID	MA (Years)	GA (Weeks)	Ultrasound Findings	CMA Findings	Type of CNV	Size (Mb)	Interpretation	Outcomes
P8245	25.5	35.1	Upper digestive obstruction	arr17q12(34822465–36410559) × 3	Duplication	1.59	P	TOP
A33770	32.4	25.3	DO	arr15q25.2(83198318–84834123) × 1	Deletion	1.69	P	TOP
A32309	34.0	25.9	DO, polyhydramnios	arr17q12(34822,466–36397279) × 3	Duplication	1.57	P	TOP
A29186	29.0	30.9	DO	arr7q11.23(72718277–74143060) × 1	Deletion	1.42	P	TOP
A26285	28.8	26.4	DO	arr13q21.33q31.1(70986515–80427919) × 1	Deletion	9.44	P	TOP
A12873	27.0	23.9	DO	arr13q21.2q31.1(60952932–84894325) × 1	Deletion	23.94	P	TOP
A30328	19.3	18.7	Microgastria, polyhydramnios, omphalocele	arr16p13.11p12.3(15375912–18778064) × 1	Deletion	3.40	P	Neonatal death
A29609	35.5	26.3	gallbladder atresia, polycystic kidney	arr17q12(34822465–36418529) × 1	Deletion	1.60	P	TOP
R4	28.4	39.0	Peritoneal effusion	arrXq28(154952431–155233098) × 3	Duplication	2.52	P	TOP
				arrYp11.31q11.23(2650424–28799654) × 2		26.15		
A34806	30.9	28.0	Intestinal echo enhancement, FGR	arrXp22.31(6455152–8135568) × 1	Deletion	1.68	P	TOP
A31334	25.3	27.1	Intestinal echo enhancement	arr(X) × 2, (Y) × 1	Duplication	155.07	P	Live birth

DSM: digestive system malformations; CMA: chromosomal microarray analysis; MA: maternal age; GA: gestational age; CNV: copy number variant; P: pathogenic; TOP: termination of pregnancy; DO: duodenal obstruction; FGR: fetal growth restriction.

**Table 2 genes-14-01872-t002:** Diagnostic genetic variants identified in positive diagnostic cases.

Case ID	Patient ID	Digestive System Abnormality	Other Ultrasound Abnormalities	Gene	Reference Sequence	Nucleotide/Protein Change	Whether gnomAD and In-House Databases Were Documented	Inheritance and ACMG Classification	Disorders (OMIM IDs)	Outcome
1	P8932	Ascites	Cerebellar vermis missing, polydactyly deformity	*IFT74*	NM_025103.4	c.535C > G (p. Gln179Glu)	Yes (0.001406 *)/yes (0.003 ^♯^)	Het, Pat, AR, P (PVS1 + PS2 + P M2)	Joubert syndrome 40 (619582), Bardet–Biedl syndrome 22 (617119)	Live birth
						c.909dup (p. Glu304ArgfsTer7) ＊	Yes (0.00005674 *)/no	Het, Mat, AR, P (PVS1 + PS2 + P M2)		
2	A35570	Hepatomegaly	Polyhydramnios, Short femur	*COL2A1*	NM_001844.5	c.1835G > T (p.Gly612Val) ＊	No/no	Het, De novo, AD, LP (PVS1 + PM2)	Spondyloepiphyseal dysplasia congenita (183900)	TOP
3	A35151	Non-visualization of the gallbladder	Short femur	*SHOX*	NM_000451.4	Loss of exons 2–6 (copy number, X1) ＊	No/no	Het, De novo, AD, P (PVS1 + PS2 + PM2)	Leri–Weill dyschondrosteosis (127300)	Live birth
4	A33669	Microgastria	Bilateral foot varus, duplication of kidney, polyhydramnios	*SMPD4*	NM_017951.5	c.1867del (p. Leu623TrpfsTer130) ＊	No/no	Het, Mat, AR, P (PVS1 + PM2 + PP4)	Neurodevelopmental disorder with microcephaly, arthrogryposis, and structural brain anomalies (618622)	TOP
						c.2074G > T (p. Glu692Ter)	No/no	Het, Pat, AR, LP (PVS1 + PM2)		
5	A34905	Duodenal obstruction	Omphalocele	*PTPN11*	NM_002834.5	c.867G > T (p. Arg289Ser)	Yes (0.000003978 *)/no	Het, De novo, AD, LP (PVS1 + PM2)	Noonan syndrome 1(163950)/LEOPARD syndrome 1 (151100)	Live birth
6	A34170	Microgastria	Polyhydramnios	*EFTUD2*	NM_004247.4	c.623_624del (p. His208ArgfsTer5) ＊	No/no	Het, De novo, AD, P (PP1_very strong + PM1 + PP3)	Mandibulofacial dysostosis, Guion–Almeida type (610536)	Live birth
7	P9429	Intestinal dilatation	Microcephaly	*17p13.3*	-	3.51 Mb deletion	No/no	De novo, AD, P (PVS1 + PM2 + PP4)	Miller–Dieker lissencephaly syndrome (247200)	TOP
8	A32697	Duodenal obstruction	Bilateral lateral ventricle widened	*17q12*	-	1.49 Mb microduplication	No/no	Mat, AD, P (PVS1 + PM2 + PP4)	Chromosome 17q12 duplication syndrome (614526)	TOP
9	C7699	Ascites	Abnormal foot posture, cardiovascular malformations	*CPLANE1*	NM_023073.3	c.1270C > T (p. Arg424Ter)	Yes (0.0001757 *)/no	Het, Pat, AR, P (PVS1 + PS2 + PM2)	Orofaciodigital syndrome VI (277170) Joubert syndrome 17 (614615)	TOP
						c.4034A > G (p. Gln1345Arg)	No/no	Het, Mat, AR, LP (PVS1 + PM2)		
10	P9381	Abdominal mass	Hydronephrosis	*OTOF*	NM_194248.2	c.897 + 1G > T	Yes (0.00004619 *)/no	Het, Pat, AR, LP (PVS1 + PM2)	Deafness, autosomal recessive 9 (601071)	Live birth
						c.4023 + 1G > A	Yes (0.01178 *)/yes (0.0089 ^♯^)	Het, Mat, AR, VUS (PM1 + PP3)		
11	NG1131	Microgastria, esophagotracheal fistula	Hydronephrosis, FGR	*NIPBL*	NM_133433.3	c.6983C > A (p. Thr2328Lys) ＊	No/no	Het, De novo, AD, LP (PS2 + PM1 + PM2 + PP3)	Cornelia de Lange syndrome 1 (122470)	TOP
12	P9325	Ascites	No amniotic fluid	*NPC1*	NM_000271.4	c.2526T > A (p. Phe842 Leu) ＊	No/no	Het, Mat, AR, VUS (PVS1 + PM2_supportin)	Niemann–Pick disease, type C1 (257220)	TOP
						c.1226T > C (p. Ile409Thr)	No/no	Het, De novo, AR, LP (PVS1 + PM2)		
13	C7430	Ascites	NT thickening, cardiovascular malformations	*FLNA*	NM_001110556.1	c.1538_ 1539delGGinsTA (p. Gly513Val) ＊	No/no	Hemi, Mat, XLD/XLR, LP (PVS1 + PM2)	Cardiac valvular dysplasia, X-linked (314400), congenital short bowel syndrome (300048)	TOP
14	A31257	Ascites	-	*HRAS*	NM_005343.4	c.38G > A (p. Gly13Asp)	No/no	Het, De novo, AD, P (PVS1 + PS2 + PM2)	Costello syndrome (218040), thyroid cancer, nonmedullary, 2 (188470)	TOP
15	P9203	Abnormality of the intrahepatic bile duct	Cardiovascular malformations, polycystic kidney, hyperechogenic kidney, no amniotic fluid	*PKHD1*	NM_138694.3	c.107C > T (p. Thr36Met)	Yes (0.0005094 *)/yes (0.001 ^♯^)	Het, Mat, AR, P (PVS1 + PS2 + PM2)	Polycystic kidney disease 4, with or without hepatic disease (263200)	TOP
						c.5869G > A (p. Asp1957Asn)	Yes (0.00005438 *)/ yes (0.001 ^♯^)	Het, Pat, AR, LP (PVS1 + PM2)		
16	W6488	Ascites	Pleural effusion, hydrops fetalis	*PTPN11*	NM_002834.5	c.1507G > A (p. Gly503Arg)	No/no	Het, De novo, AD, P (PS2 + PM1 + P M2 + PP3)	Noonan syndrome 1(163950)/ LEOPARD syndrome 1 (151100)	TOP
17	NG800	Ascites	FGR, hydrops fetalis, oligohydramnios, VSD	*KRAS*	NM_033360.4	c.101C > T (p. Pro34Leu)	No/no	Het, De novo, AD, P (PVS1 + PS2 + PM2)	Noonan syndrome 3 (609942), cardiofaciocutaneous syndrome 2 (615278)	Live birth
18	C6196	Hepatomegaly	Omphalocele, polyhydramnios, short long bone	*CDKN1C*	NM_000076.2	c.827_ 828delinsAA (p. F276X) ＊	No/no	Het, Mat, AD, LP (PM2 + PM3 + PM4)	Beckwith–Wiedemann syndrome (130650), IMAGE syndrome (614732)	Live birth
19	A28929	Hepatomegaly	Omphalocele, urachal cyst, hydrocephalus, hyperechogenic kidneys	*CRB2*	NM_173689.6	c.3548T > A (p. L1183X) ＊	No/no	Het, Pat, AR, P (PVS1 + PM2 + PP4)	Ventriculomegaly with cystic kidney disease (219730), focal segmental glomerulosclerosis-9 (616220)	Live birth
						c.3307T > C (p. C1103R) ＊	No/no	Het, Mat, AR, LP (PM2 + PM3 + PM4)		
20	P8465	Ascites	Multicystic kidney dysplasia	*BBS2*	NM_031885	c.534 + 1G > T	No/no	Het, Mat, AR, LP (PVS1 + PM2)	Bardet–Biedl syndrome-2 (615981)	Neonatal death
						c.2107C > T (p.R703X)	Yes (0.0003 *)/yes (0.001 ^♯^)	Het, Pat, AR, P (PS2 + PM1 + PM2 + PP3)		
21	A25666	Gastrointestinal obstruction	Situs inversus totalis, dextrocardia, abnormal vena cava morphology, ventriculomegaly	*DNAH11*	NM_001277115.1	c.11392G > T (p. Glu3798Ter) ＊	No/no	Het, Pat, AR, LP (PS2 + PM1 + PM2 + PP3)	Ciliary dyskinesia, primary, 7, with or without situs inversus (611884)	TOP
						c.11374-18A > G ＊	No/no	Het, Mat, AR, VUS (PVS1 + PM2_supporting)		
22	A23948	Ascites, hepatomegaly	-	*PYGL*	NM_002863	c.772 + 1G > A	No/no	Hom, Mat, AR, LP (PVS1 + PM2)	Glycogen storage disease VI (232700)	Live birth
23	A33772	Ascites, abnormal intestine morphology	Ureteropelvic junction obstruction, cloacal abnormality, renal duplication	*22q11.2*	-	2.85 Mb deletion	No/no	De novo, AD, P (PS2 + PS3 + PM2 + PP3)	Chromosome 22q11.2 deletion syndrome (611867)	TOP

＊: the first reported variant; *: allele frequencies of East Asian populations; ^#^: allele frequencies of Chinese populations; Hemi: hemizygous; Mat: maternal-inherited; XLR: X-linked recessive; P: pathogenic; TOP: termination of pregnancy; Het: heterozygous; Hom: homozygous; AD: autosomal dominant; LP: likely pathogenic; Pat: paternal-inherited; AR: autosomal recessive; VUS: variants of unknown significance; XLD: X-linked dominant; NT: nuchal translucency; IMAGE: intrauterine growth retardation, metaphyseal dysplasia, adrenal hypoplasia congenita, and genital anomalies; PD: pseudoautosomal dominant inheritance (refers to the genetic mode caused by the conventional exchange of pseudoautosomal region genes on X and Y chromosomes).

**Table 3 genes-14-01872-t003:** Diagnostic rate in different types of digestive system malformations.

DSM Classification	Isolated DSM		Nonisolated DSM		Total	
	N	Diagnostic Rate (%)	N	Diagnostic Rate (%)	N	Diagnostic Rate (%)
Abnormal duodenum morphology	15	0 (0)	26	2 (7.7)	41	2 (4.9)
Absent gallbladder	6	0 (0)	8	1 (12.5)	14	1 (7.1)
Ascites	6	1 (16.7)	26	9 (34.6)	32	10 (31.25)
Hepatomegaly	5	1 (20.0)	13	3 (23.1)	18	4 (22.2)
Abnormal stomach bubble	5	0 (0)	11	3 (27.3)	16	3 (18.75)
Abdominal mass	1	0 (0)	3	1 (33.33)	4	1 (25.0)
Others	8	0 (0)	10	2 (20)	18	2 (11.1)
Total	46	2 (4.3)	97	21 (21.6)	143	23 (16.1)

DSM: digestive system malformation.

**Table 4 genes-14-01872-t004:** Statistical analysis of the genetic and clinical outcomes of DSM fetuses.

Groups	CMA			ES		
	P/LP	VUS	Live Birth	P/LP	VUS	Live Birth
Isolated	7 (1.4%)	23 (4.7%)	354 (72.0%)	2 (4.3%)	1 (2.2%)	35 (76.1%)
Nonisolated	4 (3.9%)	46 (44.7%)	37 (35.9%)	21 (21.6%)	26 (26.8%)	47 (48.5%)
*p*-value	0.105	0.000	0.000	0.007	0.000	0.002

DSM: digestive system malformation; CMA: chromosomal microarray analysis; ES: exome sequencing; P: pathogenic; LP: likely pathogenic; VUS: variants of unknown significance.

## Data Availability

The data that support the findings of this study are not publicly available as the information contained could compromise the privacy of research participants. Further inquiries can be directed to the corresponding author.

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
