# Peer review of "Chromosome Microarray Analysis and Exome Sequencing: Implementation in Prenatal Diagnosis of Fetuses with Digestive System Malformations"

_genes, 2023, doi:10.3390/genes14101872_

Round 1

Reviewer 1 Report

Congratulations on this very interesting and innovative paper. 

I just have a few remarks:

- line 188: how exactly did you diagnose meconium peritonitis on prenatal ultrasound?

- line 297 table 3: this table is very crowded, please improve the lay-out

- line 303 figure 1: please explain the used abbreviations in the figure to improve readability

- line 324: please comment: what would be the advantage of CMA + karyotype vs. CMA only

- line 338 ff: you can correct for this shortcoming by using HPO terms regarding the postnatal phenotype and compare to the prenatal ultrasound findings

Author Response

Dear Editor and Reviewer 1,

Thank you very much for giving us an opportunity to revise our manuscript entitled, " Chromosome microarray analysis and exome sequencing: im-plementation in prenatal diagnosis of fetuses with digestive system malformations" Genes-2608632. We appreciate the time and effort that you and the reviewers dedicated to providing feedback on our manuscript and are grateful for the insightful comments on and valuable improvements to our paper. We have carefully studied the reviewer's comments carefully and tried our best to revise according to the comments. The language editing is carried out by an English expert. The English editing certificate has been uploaded. Revised portions are marked in red in the revised paper.

We have incorporated most of the suggestions made by the reviewers. Those changes are highlighted within the manuscript. Please see below for a point-by-point response to the reviewer's comments and concerns. All page numbers refer to the revised manuscript file with tracked changes.

Thank you very much for your attention and consideration.

We would like also to thank you for allowing us to resubmit a revised copy of the manuscript.

We hope that the revised manuscript is accepted for publication in Genes.

Institute: The first clinical medical college, Southern Medical University, Guangzhou, China

Tel: +86 (0)20 38076346

Fax: +86 (0)20 38076337

E-mail: You Wang, wy13781539630@163.com; Can Liao, canliao6008@163.com

Sincerely yours

Dr. You Wang

Response to Reviewer 1 Comments:

Congratulations on this very interesting and innovative paper.

I just have a few remarks:

- line 188: how exactly did you diagnose meconium peritonitis on prenatal ultrasound?

- line 297 table 3: this table is very crowded, please improve the lay-out

- line 303 figure 1: please explain the used abbreviations in the figure to improve readability

- line 324: please comment: what would be the advantage of CMA + karyotype vs. CMA only

- line 338 ff: you can correct for this shortcoming by using HPO terms regarding the postnatal phenotype and compare to the prenatal ultrasound findings

Thank you very much for your interest in our manuscript. In future research, we will strive to write more meaningful articles.

Each of your comments has great significance in improving the quality of our manuscript. I am very grateful to you for this.

Point 1:

- line 188: how exactly did you diagnose meconium peritonitis on prenatal ultrasound?

Response 1:

First, thank you very much for your question. 

Regarding the ultrasound diagnosis of meconium peritonitis, our center strictly adheres to international ultrasound diagnostic guidelines. All patients had shown one or more of the following ultrasound findings (to a greater or lesser extent): fetal ascites, intra-abdominal calcification or echogenic mass, polyhydramnios, pseudocyst and dilated bowel loop. Generally speaking, intra-abdominal calcification is a characteristic manifestation of meconium peritonitis.

In order to avoid readers and you having the same confusion, we have added ultrasound diagnosis of meconium peritonitis in the manuscript in lines 192-194.

Your question has made our manuscript more rigorous, and I would like to thank you for that.

I hope our answer can satisfy you.

Point 2:

- line 297 table 3: this table is very crowded, please improve the lay-out

Response 2:

Thank you very much for your comment. We have improved the layout of Table 3 in the newly revised manuscript version. 

Your comments have made the layout of our manuscript more aesthetically pleasing, and I would like to thank you for that.

Point 3:

- line 303 figure 1: please explain the used abbreviations in the figure to improve readability

Response 3:

Thank you very much for your very constructive suggestions. We have explained all the abbreviations in Figure 1 in the newly revised manuscript version.

Your suggestion has made our manuscript more rigorous, and I would like to express my gratitude to you for this.

Point 4:

- line 324: please comment: what would be the advantage of CMA + karyotype vs. CMA only

Response 4:

Thank you very much for your constructive suggestions. We have added the advantages of CMA+ karyotype vs. CMA only in the newly revised manuscript version:

“In addition, the advantages of implementing CMA+karyotype compared to only implementing CMA are when routine chromosome karyotype analysis is performed for any indication, the detection rate of abnormalities is 0.6% higher than CMA. However, as we and others have shown, most of this rate can be explained by balanced inversions and translocations. Triploids can also be detected through routine chromosome karyotype analysis but missed diagnosis by CMA.” in lines 368-374. We have also cited corresponding references.

Your suggestion has made our manuscript more rigorous, and I am very grateful for that.

Point 5:

- line 338 ff: you can correct for this shortcoming by using HPO terms regarding the postnatal phenotype and compare to the prenatal ultrasound findings

Response 5:

Thank you very much for your constructive suggestions.

Based on your suggestion, we have further supplemented our manuscript.We followed up the patient again and recorded the ultrasound phenotypes of the postpartum ultrasound examination in detail, and compared them with the findings of prenatal ultrasound.In the newly revised manuscript version, we have added relevant information in the results section: "Among the live births, 334 (85.42%) received postpartum ultrasound reexamination, and 57 (14.58%) did not. We compared the postpartum phenotype with prenatal ultrasound findings by using HPO terms. 308 cases (78.77%) were found to be consistent with prenatal and postpartum ultrasound findings. 26 cases (6.65%) showed new phenotypes in post-partum ultrasound scans, including 7 cases of atrial septal defect, 5 cases of ventricular septal defect, 5 cases of renal cyst, 4 cases of biliary atresia, 4 cases of congenital clubfoot and 1 case of renal agenesis." in lines 278-284.

In addition, we have also made modifications to the relevant content of the discussion section.

The original sentence has been modified to: “It is worth noting that the classification of isolated and non-isolated DSM was based on the results of prenatal ultrasonography in this study. Although we have corrected this shortcoming by using HPO terms regarding the postnatal phenotype and comparing it to the prenatal ultrasound findings, in fact, fetal phenotypes that can be identified using prenatal ultrasound are limited. For example, mental retardation may be unrecognizable on prenatal ultrasonography. These objective factors complicate the prenatal ES analysis.”in lines 386-392.

Sincerely hope that our modifications can meet your satisfaction. Your suggestion has made our manuscript more comprehensive. Thank you again for this.

Reviewer 2 Report

Thank you for sharing your great experience on this large series. This is of great value to the community of prenatal diagnosis practitioners.

It is particularly interesting to read about your experience with VUS, not all prenatal diagnostic centers (PDC) deliver these results to parents. Your PDC does it and the results show that VUS are frequent; they influence the choice of parents regarding the use of TOP, especially when they are de novo. Additionally, the neurodevelopmental assessment in you provide in live birth is not completely reassuring. "Our study also showed that the incidence of mental retardation in children with DSM with VUS was significantly higher than in those with negative results (18.0% versus 8.0%, p=0.010) )”.

The question that immediately comes to mind is, are these results sufficiently substantiated and unbiased to be published?

My answer is neither black nor white it is grey: The condition for publication is that you must mention in your conclusion that the results on the real meaning of VUS must be confirmed by other studies.

Apart from this main question I have some criticisms :

lines 81-82 I didn't understand if you consider the DSMs with ultrasound soft markers isolated? please make your sentence clearer.

Ditto for lines 213 - 224 the results are messy and sometimes erroneous

The presentation of Table 4 needs to be improved

The flowchart in Figure 1 is inconsistent and needs to be revised

Author Response

Dear Editor and Reviewer 2,

Thank you very much for giving us an opportunity to revise our manuscript entitled, " Chromosome microarray analysis and exome sequencing: implementation in prenatal diagnosis of fetuses with digestive system malformations" Genes-2608632. We appreciate the time and effort that you and the reviewers dedicated to providing feedback on our manuscript and are grateful for the insightful comments on and valuable improvements to our paper. We have carefully studied the reviewer's comments carefully and tried our best to revise according to the comments. The language editing is carried out by an English expert. The English editing certificate has been uploaded. Revised portions are marked in red in the revised paper.

We have incorporated most of the suggestions made by the reviewers. Those changes are highlighted within the manuscript. Please see below for a point-by-point response to the reviewer's comments and concerns. All page numbers refer to the revised manuscript file with tracked changes.

Thank you very much for your attention and consideration.

We would like also to thank you for allowing us to resubmit a revised copy of the manuscript.

We hope that the revised manuscript is accepted for publication in Genes.

Institute: The first clinical medical college, Southern Medical University, Guangzhou, China

Tel: +86 (0)20 38076346

Fax: +86 (0)20 38076337

E-mail: You Wang, wy13781539630@163.com; Can Liao, canliao6008@163.com

Sincerely yours

Dr. You Wang

Response to Reviewer 2 Comments:

Thank you for sharing your great experience on this large series. This is of great value to the community of prenatal diagnosis practitioners.

It is particularly interesting to read about your experience with VUS, not all prenatal diagnostic centers (PDC) deliver these results to parents. Your PDC does it and the results show that VUS are frequent; they influence the choice of parents regarding the use of TOP, especially when they are de novo. Additionally, the neurodevelopmental assessment in you provide in live birth is not completely reassuring. "Our study also showed that the incidence of mental retardation in children with DSM with VUS was significantly higher than in those with negative results (18.0% versus 8.0%, p=0.010) )”.

The question that immediately comes to mind is, are these results sufficiently substantiated and unbiased to be published?

My answer is neither black nor white it is grey: The condition for publication is that you must mention in your conclusion that the results on the real meaning of VUS must be confirmed by other studies.

Apart from this main question I have some criticisms :

lines 81-82 I didn't understand if you consider the DSMs with ultrasound soft markers isolated? please make your sentence clearer.

Ditto for lines 213 - 224 the results are messy and sometimes erroneous

The presentation of Table 4 needs to be improved

The flowchart in Figure 1 is inconsistent and needs to be revised

Point 1:

Thank you for sharing your great experience on this large series. This is of great value to the community of prenatal diagnosis practitioners.

It is particularly interesting to read about your experience with VUS, not all prenatal diagnostic centers (PDC) deliver these results to parents. Your PDC does it and the results show that VUS are frequent; they influence the choice of parents regarding the use of TOP, especially when they are de novo. Additionally, the neurodevelopmental assessment in you provide in live birth is not completely reassuring. "Our study also showed that the incidence of mental retardation in children with DSM with VUS was significantly higher than in those with negative results (18.0% versus 8.0%, p=0.010) )”.

The question that immediately comes to mind is, are these results sufficiently substantiated and unbiased to be published?

My answer is neither black nor white it is grey: The condition for publication is that you must mention in your conclusion that the results on the real meaning of VUS must be confirmed by other studies.

Response 1:

Dear reviewer, first of all, thank you very much for your approval of our manuscript.

Based on your highly constructive suggestions, we have made modifications in our newly revised manuscript.

We have added this shortcoming to the limitations section “Furthermore, although we mentioned in our study that the incidence of mental retar-dation in DSM children with VUS is significantly higher than those with negative results, we cannot deny that the true significance of VUS must be confirmed by other studies. ” in lines 483-485.

Thank you very much for your suggestion, which has made our article more rigorous.

Point 2:

lines 81-82 I didn't understand if you consider the DSMs with ultrasound soft markers isolated? please make your sentence clearer.

Response 2:

Thank you very much for your comment. We are very sorry for our negligence. In the newly revised manuscript version, we have revised the original sentence to read: “In this study, the fetus was classified into the isolated group, regardless of whether the fetus has ultrasound soft markers or not. ”in line 82.

Your comments make our manuscript more rigorous, for which I am very grateful to you.

Point 3:

Ditto for lines 213 - 224 the results are messy and sometimes erroneous

Response 3:

Thank you very much for your comment.

I am very sorry for our lack of language expression skills.

We have revised the original sentence in the newly revised version of the manuscript.

In the newly revised manuscript version, we have revised the original sentence to read: “Overall, ES detected 29 P/LP variants and three cases of microdeletions and microdu-plications in 23 cases (Table 2) and 26 VUS in 26 cases (Table S2). Of the 29 diagnostic variants identified, 14 variants were not reported previously. The diagnostic yield of ES was 16.08% (23/143).

In the 23 positive cases, de novo variants were identified in 11 cases. In these 11 cases, 10 cases (90.9%) were autosomal dominant, and one case (9.1%) was autosomal recessive. Compound heterozygotes or homozygotes were found in 10 cases with autosomal re-cessive inheritance. One hemizygote was inherited from his mother with X-linked dominant/recessive inheritance. The remaining fetus was inherited from his mother without a clinical phenotype with autosomal dominant.” in lines 218-227.

Your comments have made our manuscript clearer and more accurate, and I am very grateful for that.

Point 4:

The presentation of Table 4 needs to be improved

Response 4:

Thank you very much for your constructive suggestions.

We have made modifications to the presentation of Table 4 in the newly revised manuscript version.

I hope our modification this time can meet your satisfaction. If there are still shortcomings, please let me know without hesitation. Thank you.

Your suggestion has made our manuscript more aesthetically pleasing, and for this, I would like to thank you again.

Point 5:

The flowchart in Figure 1 is inconsistent and needs to be revised

Response 5:

Thank you very much for your suggestion. I am very sorry for our negligence.

In the newly revised manuscript version, we have made modifications to Figure 1.

Your suggestion has made our manuscript more accurate, and for this, I would like to thank you again.

Round 2

Reviewer 2 Report

Thank you very much, the text is now very clear and useful for the community of prénatal diagnosis practitioner